# Affective states in digital game-based learning: Thematic evolution and social network analysis

**Xieling Chen[1], Di Zou[2]\*, Lucas Kohnke[2], Haoran Xie[3], Gary Cheng[1]**

**1** Department of Mathematics and Information Technology, The Education University of Hong Kong, Hong Kong, Hong Kong SAR, **2** Department of English Language Education, The Education University of Hong Kong, Hong Kong, Hong Kong SAR, **3** Department of Computing and Decision Sciences, Lingnan University, Hong Kong, Hong Kong SAR

\* dizoudaisy@gmail.com

**Data Availability Statement:** The data used in this study is provided in the Supporting information.

**Funding:** Haoran Xie's work is supported by Direct Grant (DR21A5) and the Faculty Research Grant (DB21A9) of Lingnan University, Hong Kong. Gary

## Abstract

Research has indicated strong relationships between learners' affect and their learning. Emotions relate closely to students' well-being, learning quality, productivity, and interaction. Digital game-based learning (DGBL) has been widely recognized to be effective in enhancing learning experiences and increasing student motivation. The field of emotions in DGBL has become an active research field with accumulated literature available, which calls for a comprehensive understanding of the up-to-date literature concerning emotions in virtual DGBL among students at all educational levels. Based on 393 research articles collected from the Web of Science, this study, for the first time, explores the current advances and topics in this field. Specifically, thematic evolution analysis is conducted to explore the evolution of topics that are categorized into four different groups (i.e., games, emotions, applications, and analytical technologies) in the corpus. Social network analysis explores the co-occurrences between topics to identify their relationships. Interesting results are obtained. For example, with the integration of diverse applications (e.g., mobiles) and analytical technologies (e.g., learning analytics and affective computing), increasing types of affective states, socio-emotional factors, and digital games are investigated. Additionally, implications for future research include 1) children's anxiety/attitude and engagement in collaborative gameplay, 2) individual personalities and characteristics for personalized support, 3) emotion dynamics, 4) multimodal data use, 5) game customization, 6) balance between learners' skill levels and game challenge as well as rewards and learning anxiety.

## Introduction

Digital games are well-recognized as effective tools to provide intrinsically motivating experiences with heightened enjoyment of doing "for its own sake" [1]. Accumulative evidence shows digital games' potentials to nurture and sustain high levels of learning motivation and engagement [2]. Digital game-based learning (DGBL) is thus increasingly recognized as an

Cheng's work is supported by One-off Special Fund from Central and Faculty Fund in Support of Research from 2019/20 to 2021/22 (MIT02/19-20), Research Cluster Fund (RG 78/2019-2020R), and Dean's Research Fund 2019/20 (IDS-2 2020) of The Education University of Hong Kong. The funders had no role in study design, data collection and analysis, decision to publish, or preparation of the manuscript.

**Competing interests:** Di Zou and Haoran Xie are currently the academic editors of PLOS ONE. This does not alter our adherence to PLOS ONE policies on sharing data and materials.

invaluable medium to promote emotionally engaging learning experiences [3]. In addition to engaging and entertaining learners, DGBL helps to develop learners' cognitive abilities, encourage problem-solving, facilitate collaborations, and raise self-esteem [4]. Understanding the affective status of students at all educational levels during virtual DGBL is thus essential for developing suitable learning strategies and improving DGBL learning gains and achievements. Consequently, DGBL's combination with learner affective status has developed into a field increasingly and actively researched in both academia and educational sectors.

DGBL typically elicits a wide range of emotions that are subjective, complex, and difficult to grasp but are fundamental to facilitate effective instruction and learning [5]. Research into emotions and emotional engagement suggest emotions' importance in facilitating DGBL. Accordingly, increasing attention is paid to various emotions (e.g., frustration and joy) that arise during DGBL [6] and their potentials to facilitate or hinder learning [7]. The body of literature on learners' affective status during DGBL is expanding. Nevertheless, few reviews on emotions in DGBL have been conducted, with only two relevant reviews available. Specifically, Henritius et al. [8] systematically reviewed articles concerning university students' emotions in virtual learning during 2002–2017, and Abdul Jabbar and Felicia [9] systematically analyzed game design features that promoted engagement and learning in DGBL during 2003–2013. The scopes (i.e., emotions in virtual learning and engagement in DGBL) of the two reviews are not exactly similar to ours (i.e., emotions in DGBL), thus restricting the analyses of the types of emotions and digital games as well as technologies and devices applied to support DGBL. Additionally, we do not have a clear understanding of what types of topics (i.e., keywords that are extracted from titles and abstracts of the analyzed articles) have been investigated and how they evolve and correlate. Consequently, there is no in-depth review that summarizes the state-of-the-art literature concerning emotions in DGBL and analyzing how diverse types of emotions and digital games are linked and how diverse technologies and devices have been used to support DGBL by triggering positive emotions while reducing negative ones.

Given the importance of learners' affective status in DGBL, either as influencing factors affecting DGBL's effectiveness or as learning outcomes resulted from DGBL, a review study that brings together the fields of DGBL and emotions to summarize how learners' affective status are concerned and studied in DGBL literature, appears essential. Such analyses enable scholars and instructors to better understand what, how, and to what extent different types of emotions are concerned and addressed in DGBL, particularly with the use of what types of analytical technologies, which may be useful for them when making decisions about designing research studies and planning instructional activities.

More specifically, this bibliometric study provides an objective overview of the affective status of students at all educational levels in virtual DGBL based on 393 research articles indexed in the Science Citation Index Extended (SCIE) and the Social Science Citation Index (SSCI) databases. There are three research questions.

RQ 1: What are the important topics related to emotions, games, applications, and analytical technologies?

RQ2: How do these topics evolve?

RQ3: How do these topics correlate?

To answer the above questions, thematic evolution analysis and social network analysis are conducted. Specifically, as for RQ1, we identify important topics related to emotions, games, applications, and analytical technologies described in the titles and abstracts of the 393 articles. Then, thematic evolution analysis explores the evolution of topics related to games, emotions, applications, and analytical technologies to answer RQ2. To enable a detailed understanding

of how topics evolve, we divide the whole period into three sub-periods, each consisting of two years, i.e., 2014–2015 (92 articles), 2016–2017 (132 articles), and 2018–2019 (169 articles). A two-year sub-period is suitable by considering the balance in the numbers of articles in the three sub-periods. Social network analysis is used to answer RQ3 by exploring the relationships between topics to understand how emotions are considered in various types of games supported by different devices and analytical technologies.

The organization of this study is as follows. After the introduction section is a literature review section that describes reviews on emotions in educational settings and topic analysis and topic evolution. A subsequent theoretical framework section defines and specifies the emotions, educational games, gaming approaches, methodologies, concepts, and different types of platforms supporting DGBL. Thereafter, the results section presents results about thematic evolution and co-occurrences, based on which research foci during three sub-periods are identified and presented. The dissuasion section focuses on proposing and discussing a research framework formed based on data analysis results to suggest implications for future research in this field. The conclusion section briefly summarizes this research and points out limitations and future efforts.

## Literature reviews

### Reviews on emotions in educational settings

There are reviews concerning emotions in educational settings. Henritius et al. [8] systematically reviewed 91 articles on college students' emotions in virtual learning environments during 2002–2017, focusing on concepts regarding emotions, contextual factors, methodologies, and findings. With a focus on learners' engagement during DGBL, Abdul Jabbar and Felicia [9] systematically analyzed game and gaming characteristics promoting engagement and learning in DGBL based on 91 articles during 2003–2013. They highlighted the importance of complex gaming elements that impacted gameplay engagement and learning experiences in DGBL at both cognitive and emotional levels. Although Henritius et al.'s review has emotions as their research target, which is similar to ours, they investigated emotions arising in virtual learning rather than in DGBL. Although with a similar target (e.g., DGBL) with ours, Abdul Jabbar and Felicia limited literature to those focusing on learners' engagement. In addition, no study focusing on research topics, topic evolutions, and topic co-occurrences in the literature concerning learners' affective status in DGBL is available. The main difference between our review and previous research is that we apply bibliometric analysis and include 393 papers for the analyses, while other studies apply systematic review methodologies and include smaller numbers of papers.

### Topic analysis and topic evolution

According to Weismayer and Pezenka [10], it is important to undertake research into the development of research fields in terms of topics and topic evolution. Rigorous bibliometric analysis is able to provide a comprehensive overview of research literature in large volume and to further derive insights not previously evaluated (e.g., [11, 12]) by allowing the identification of research topics in the past and at present quantitatively and objectively. Such analyses allow answers to questions such as "in what research topics were the scholars interested" and "how did such research topics evolve." This contributes to a deeper understanding of major topics published and how this research field has developed across the studied years [13]. In addition to identifying major topics or issues that are frequently discussed within the field, we also explore the co-occurrences between these topics to allow a better understanding of how different issues are associated and what topics are commonly simultaneously discussed. For

example, a close co-occurrence between an emotion-related topic A and an analytical technology-related topic B means that B is popularly adopted to study issues related to A. For example, as a prevalence method for studying technology acceptance, structural equation modeling is likely to be identified to closely relate to technology acceptance. Such analyses help understand hot topics across different dimensions and provide insights into potential directions by combining topics from different domains.

## Theoretical framework

To investigate and analyze the trends and developments, as well as research topics and issues concerning affective states in DGBL, topics belonging to four main categories (i.e., digital games, affective status, supporting applications and devices, and analytical technologies) are considered in this study. The following paragraphs indicate how the four categories are defined.

First, digital games for promoting learning are discussed by using various concepts, for example, digital education games, serious games, DGBL, games for learning, and digital learning games. In this study, we define digital games from a broad perspective. More specifically, we focus on "digital games used within both formal and informal educational environments for the purposes of learning, teaching a particular subject or promoting engagement (p. 203) [14]." Therefore, a wide range of games (e.g., tutorial games, simulation games, role-playing games, motion-sensing games, three-dimensional virtual games, adventure games, card games, board games, and serious games) are considered. In addition to the types of digital games, we also consider game/gaming-related elements and features (e.g., collaboration, reward system, and feedback) to enable a richer insight into and comprehensive understanding of important issues covered within the literature.

Second, this study includes the category affective status to see what specific games or gaming elements triggering learners' different emotions are concerned in extant literature. According to [15], learners experiencing varied learning facilitated by diverse technologies typically undergo a wide range of positive and negative emotions (e.g., enjoyment, anxiety, and confusion). Considering diverse definitions of affective state, emotions, and their relevant terms, we adopt the definition of affective state by Scherer's [16] that contains emotions, mood, affective state, and personality trait. Besides, socio-emotions that aim at analyzing emotional aspects of social interactions among learners during group activities were also considered. We also consider relevant concepts and factors that can have direct/indirect effects on learner emotions in DGBL, for example, emotions related to subject learning (e.g., language learning anxiety), mood/affective disorders or autism that can cause negative emotions, and varied educational contexts that can cause variations in emotions.

Also, it is commonly recognized that the rising interest in DGBL is driven by technological advances with diverse devices and applications being used to support DGBL [16] and allow the realization of innovative education practices (e.g., contextual-aware DGBL and virtual reality DGBL). Thus, the exploration of applications and devices considered within the corpus helps to understand in what applications have learners' emotions been concerned and to identify potential gaps to be filled in the future. More specifically, inspired by previous reviews, we define the supporting devices and applications that run the digital game systems as different devices (e.g., [17]), for example, wearable devices (e.g., Google glasses), smartphones, tablet computers (e.g., iPad), traditional computers/devices (e.g., personal computers), and virtual reality.

In addition, technologies have expanded analytics horizons to various educational contexts, with increasingly advanced analytics technologies (e.g., text analytics, machine learning, and data mining) being utilized to enable a fine-grained analysis of diverse types and aspects of learners' affect and emotions. Thus, the exploration of analytical technologies-related topics helps understand the status of analytical technologies used in this area and identify potential technologies not been employed.

## Methods

### Formation of search terms

Before data search, we collected affective status-related terms from [8], focusing on learners' emotions during learning. These terms were categorized into three types. The first included "emotion" and terms (e.g., "affect," "feeling," and "mood") with highly overlapped meanings. The second type of emotions (e.g., "anxiety," "attitude," "enjoyment," "frustration," "joy," and "satisfaction) was developed by referring to theories on emotions in learning (e.g., social learning theories, social-cognitive models, and appraisal theories). The third included terms describing learning behaviors concerning emotions (e.g., "attachment," "engagement," "flow," and "self-regulation"). Inspired by Abdul Jabbar and Felicia [9], we used "game*" to represent various educational games and gaming approaches, methodologies, concepts (e.g., "serious games," "game-based learning," and "educational games"), and different types of platforms supporting DGBL (e.g., "video games," "computer games," "digital games," "online games," "augmented reality games," and "virtual reality games"). We also included "gamify" and "gamification" to ensure fuller data coverage.

### Data search

A search query constructed by including the above emotion- and DGBL-related terms is as TS = (("emotion" OR "affect" OR "feeling "OR "mood" OR "anxiety" OR "attitude" OR "enjoyment" OR "frustration" OR "joy" OR "satisfaction" OR "attachment" OR "engagement" OR "flow" OR "self-regulation") AND ("game*" OR "gamify" OR "gamification") AND ("learn*")). More specifically, combinations of the three sets of terms (e.g., "emotion" AND "game*" AND "learn*"; "affect" AND "game*" AND "learn*") were considered. The results were further restricted to include 1) only research articles, 2) those published between 2014 and 2019, 3) those being indexed by SCI and SSCI, and 4) those written in English. Finally, a total of 1152 research articles written in English and published between 2014 and 2019 were returned.

### Selection of eligible studies

Articles were screened to make exclusions based on 1) title check, (b) abstract, and (c) full text, according to the criteria (see Table 1) determined with reference to [8] with additional consideration of DGBL. Specifically, when deciding whether to include a paper, we started from criterion one (i.e., relevant to DGBL) to see if they explored elements and/or factors supporting

Table 1. Inclusion criteria for data screening.

| Inclusion criteria | I1 | In the context of DGBL |
|---|---|---|
| | I2 | Inclusion of measures concerning learners' affective states concepts or socio-emotional factors |
| | I3 | Empirical studies (i.e., research involving empirical evidence) |

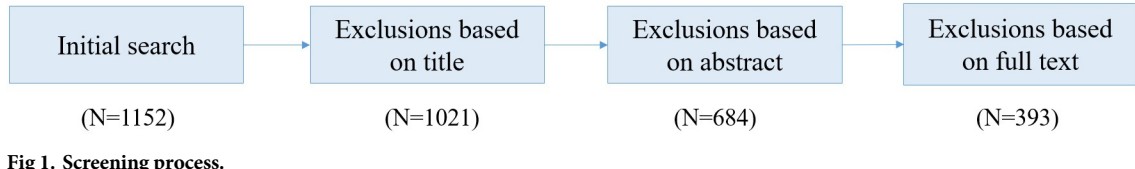

**Fig 1. Screening process.**

DGBL with empirical evidence showing their impacts on learning [9]. If it was not, we excluded it directly. In this stage, most of the excluded articles were related to physical gaming rather than learning in "virtual" games designed based on digital technologies. Also, in many studies excluded, games were purely used for entertainment without educational value. Subsequently, we read the article to check whether it involves learners' affective status, either as learning outcome measures or factors affecting learning outcomes. Many studies were excluded because they mentioned, for example, "affect," "feel," "engagement," and "self-regulation" only as broad terms, rather than specific learning outcomes or influencing factors to learning. After that, we evaluated whether a paper was an original research article and excluded reviews and survey papers. The screening resulted in 393 articles. The screening process is displayed in Fig 1. The data used in this study is available via https://drive.google.com/file/d/1bprLRhhbkYlVHnF2UN8YPjJccnbr2A-B/view?usp=sharing.

## Data extraction and analysis

Data analysis was conducted based on terms collected from titles and abstracts since they are "suitable for conceptual reviews because they usually represent the noteworthy content of articles (p. 4) [11]" and are commonly used to understand important topics and issues in literature analysis (e.g., [18]) using Natural Language Toolkit (NLTK) (https://www.nltk.org/). By providing "easy-to-use interfaces to corpora and lexical resources (e.g., WordNet) and rich text processing libraries for classification, tokenization, stemming, tagging, parsing, and semantic reasoning (p. 40) [19]," NLTK is categorized by the ability to work with human language data based on Python programs. In this study, it was mainly used for term extraction and pre-processing. Data pre-processing included number, punctuation, symbol, and stop word removal and combining singular and plural terms. However, we did not conduct synonym combinations (e.g., stemming) to diminish inflected forms and "sometimes derivationally related forms of a word to a common base form ([20], p.32)" since this might lead to difficulties in correctly interpreting word stems. For example, the stem of "organized" is "organ." Therefore, we kept the varied forms of a term.

Thematic evolution analysis explored the evolution of topics in four categories (i.e., digital games, affective status, applications and devices supporting DGBL, and analytical technologies) identified from the 393 research articles. Specifically, we identified important phrases and indicated when they emerged in the corpus by averaging the publication years of the articles where they occurred. We placed each topic in the corresponding averaged timing points. Each topic was indicated by a node with its size proportional to its occurrences.

Social network analysis, which is originated from networking and graphing theories, is popularly adopted to reveal relationships between social actors [21] displayed as nodes, with their relations being indicated by the connected links [22]. In this study, social network analysis based on topic co-occurrences explored the relationships between topics, particularly those from different categories (i.e., digital games, affective status, applications and devices supporting DGBL, and analytical technologies). Specifically, we computed the co-occurrences of two topics, that is, the number of articles in which they appeared together, and used it to measure

their relationship intensity. That is, the higher the co-occurrence value was, the more likely the two topics were to appear together in the same articles. It should be noted that we used co-occurrences rather than occurrences because co-occurrences focus on the appearance of two topics in the same articles, while occurrences indicate the appearance of a single topic in the studied articles. In the network generated using Gephi (https://gephi.org/), each topic was represented by a node with size indicating its frequency in the corpus. The connections between the nodes indicated co-occurrences, with width indicating the number of co-occurrences. Thus, the thicker the links were between topics, the more likely they were to be discussed within a paper. To enable more fine-grained illustrations and clear presentation, a series of networks involving terms with varied co-occurrence times (e.g., ranging from two to five) were drawn.

## Results

### Thematic evolution

Figs 2–4 were constructed based on Echarts (https://echarts.apache.org/zh/index.html), a powerful charting and visualization library. In the figures, topics are indicated as nodes with sizes proportional to their frequencies in the data corpus. We indicated "the timeline detailing the time of publication of the papers by the X-axis and place each topic in the corresponding timing averaged by the publication years of all the papers containing the topic ([23], p.3)." All topics with varying frequencies were considered.

Fig 2 shows the evolution of important topics related to learners' affective states. Limited emotions (e.g., satisfaction and attitude) were researched in previous years. However, with time going on, diverse concepts and issues (e.g., social engagement, English anxiety, mood disorder, affective disorder, affect regulation, and emotional trend) were studied in DGBL.

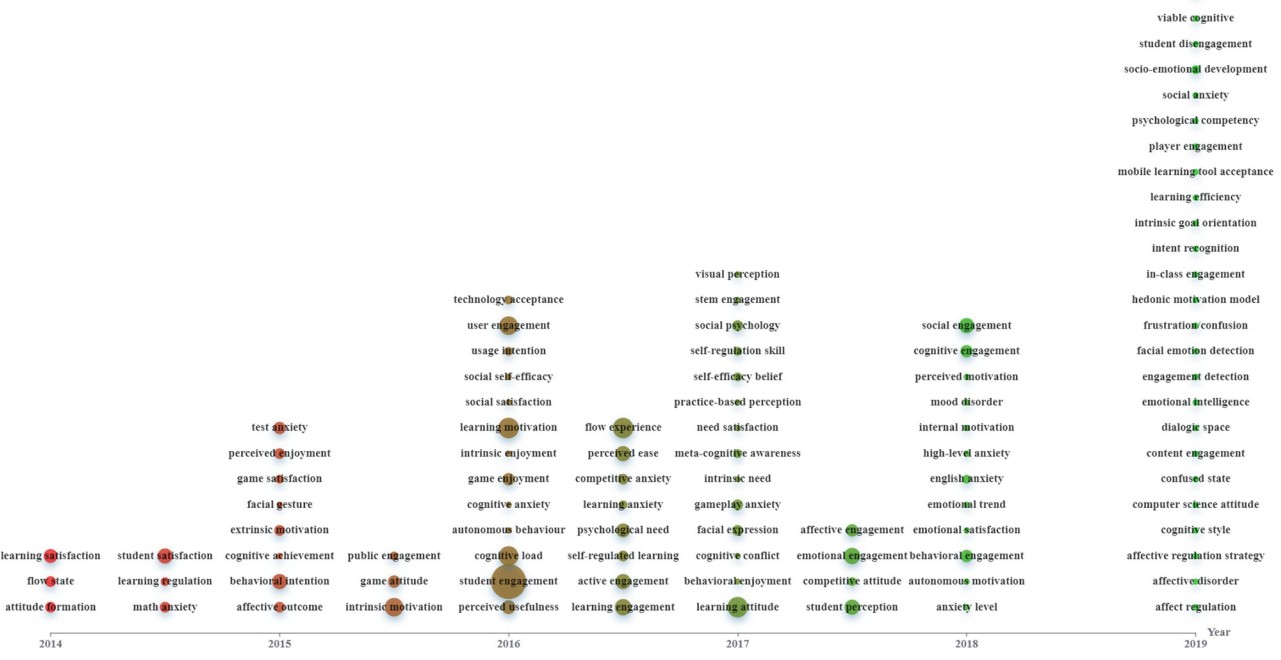

**Fig 2. Evolution of topics related to learners' affective states (download via https://drive.google.com/file/d/1rZ1JEvIIoVairoTL2JgEpOzb5xcGs1JB/view?usp=sharing).**

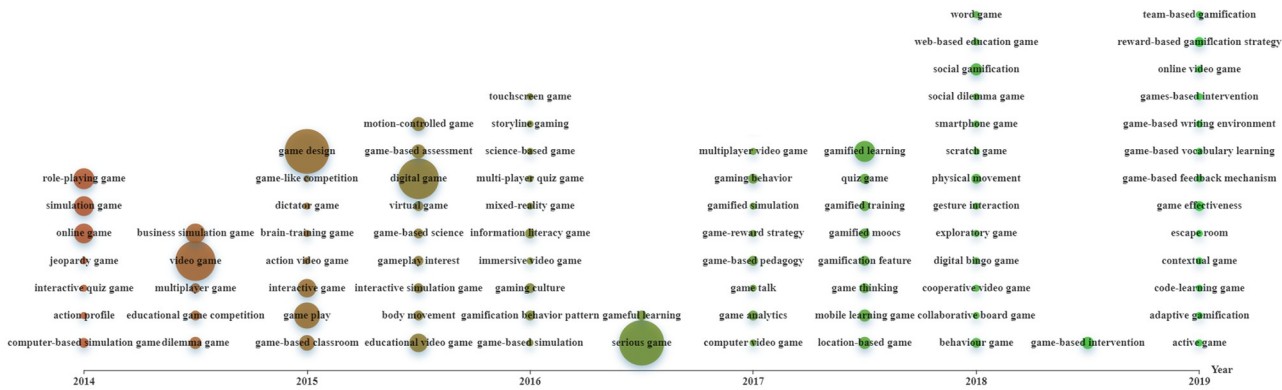

**Fig 3. Evolution of topics related to digital games and game elements (download via https://drive.google.com/file/d/1VwZ0lGWpxeFFLbsGzACa UXRSaz43GiFT/view?usp=sharing).**

Fig 3 shows the evolution of important topics related to digital games and game elements. In previous years, much attention was paid to emotions in video games. While during 2016–2017, the serious game received the most attention. In recent years, diverse issues and elements were considered, for example, exploratory games, adaptive gamification, reward strategy in DGBL, and contextual games.

Fig 4 shows the evolution of topics related to applications and devices supporting DGBL. With time going on, diverse types of applications and devices were considered. In about 2016, mobiles were increasingly used for DGBL. In 2017, social robots for childhood education were considered. In recent years, applications such as augmented reality appeared.

Fig 5 shows the evolution of analytical technologies. With time going on, diverse types of analytical technologies were used. During 2016–2017, machine learning techniques like natural language processing, clustering, relationship mining, and advanced statistical methodologies like structural equation modeling (SEM) received increasing attention. In recent years, technologies such as learning analytics, particularly game learning analytics, artificial intelligence like neural networks and affective computing started to be used.

## Thematic co-occurrences

Fig 6 shows the co-occurrences of important research topics in the four categories, with a series of network graphs depicting topics co-occurring in not less than five articles. The

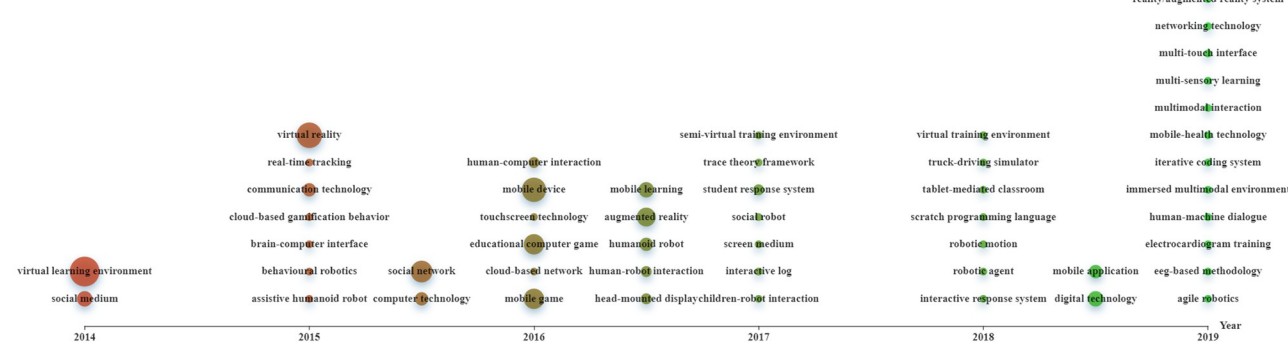

**Fig 4. Evolution of important innovative applications (download via https://drive.google.com/file/d/16Sg-zSabhNh26q4HgDkSpKdb44TRNZnv/ view?usp=sharing).**

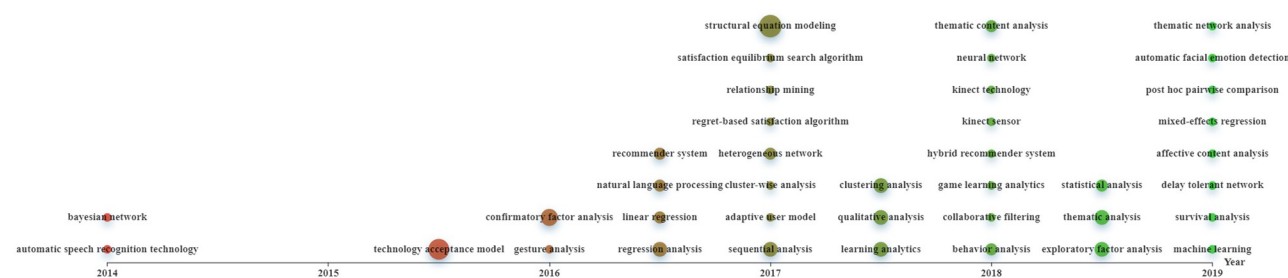

**Fig 5. Evolution of topics related to analytical technologies (download via https://drive.google.com/file/d/1cK9u2ZKP1-BWNg7y7BcaupYSMLUCp_Vn/view?usp=sharing).**

relationships between topics in categories affective states and game elements are the most significant. Fig 7 shows the topics with a co-occurrence ranging from two to four. Figs 8 and 9 show the topics linked with applications and analytical technologies with co-occurrence ranging from two to four. We can see many groups formed by topics of diverse concerns. For example, in Fig 9, a path formed by "child," humanoid robot," and "engagement" indicates a promising direction about children's engagement with humanoid robots through DGBL.

Based on the results of topic evolution and thematic co-occurrence analyses, we illustrate in the next sub-sections more detailed findings, particularly from a topic evolution perspective based on three sub-periods (i.e., 2014–2015, 2016–2017, and 2018–2019).

### Research foci during 2014–2015

During 2014–2015, the key affective status and socio-emotional factors covered in the DGBL articles were satisfaction, attitude, motivation, and particularly engagement (see Fig 2), which are also the top researched emotions identified by Henritius et al. [8]. The reason for engagement's popularity in educational game studies is because it is commonly recognized that games are mainly designed and utilized to engage learners to learn and motivate them to think and create meaning [24]. To sustain players' engagement, enjoyment and motivation are important [25]. Thus, many studies on educational games have examined the attributes and causes of enjoyment, motivation, and engagement. Although a conclusion about DGBL's positive impact on learners' learning motivation and satisfaction is commonly found, there are still studies showing the opposite side. For example, in a longitudinal study [25], learners involving in DGBL show less intrinsic motivation and satisfaction over time as compared to those in traditional classrooms. This is partially caused by the badge and coin rewards and competition and social comparison' encouragement using digital leader boards, which, according to cognitive evaluation theory, reduces intrinsic motivation since the reward is seen as controlling it, thus making learners feel less competent and in control. Such a contradictory conclusion to the majority of studies is mainly due to the difference in short- and long-term interventions. As it is common that the effect of innovations eliminates with time, it is thus essential to study gamification's long-term effects to better evaluate whether and how DGBL and specific game elements impact learners' affect and learning outcomes in the long run.

### Research foci during 2016–2017

In this period, research foci in the field became diverse, with more emotions and game-related issues being concerned. This is associated with technological advances (e.g., web 2.0, networking, and mobile technologies) (see Fig 4) that provide support to DGBL with varied elements (e.g., competition and collaboration) being considered (see Fig 3). Additionally, educators'

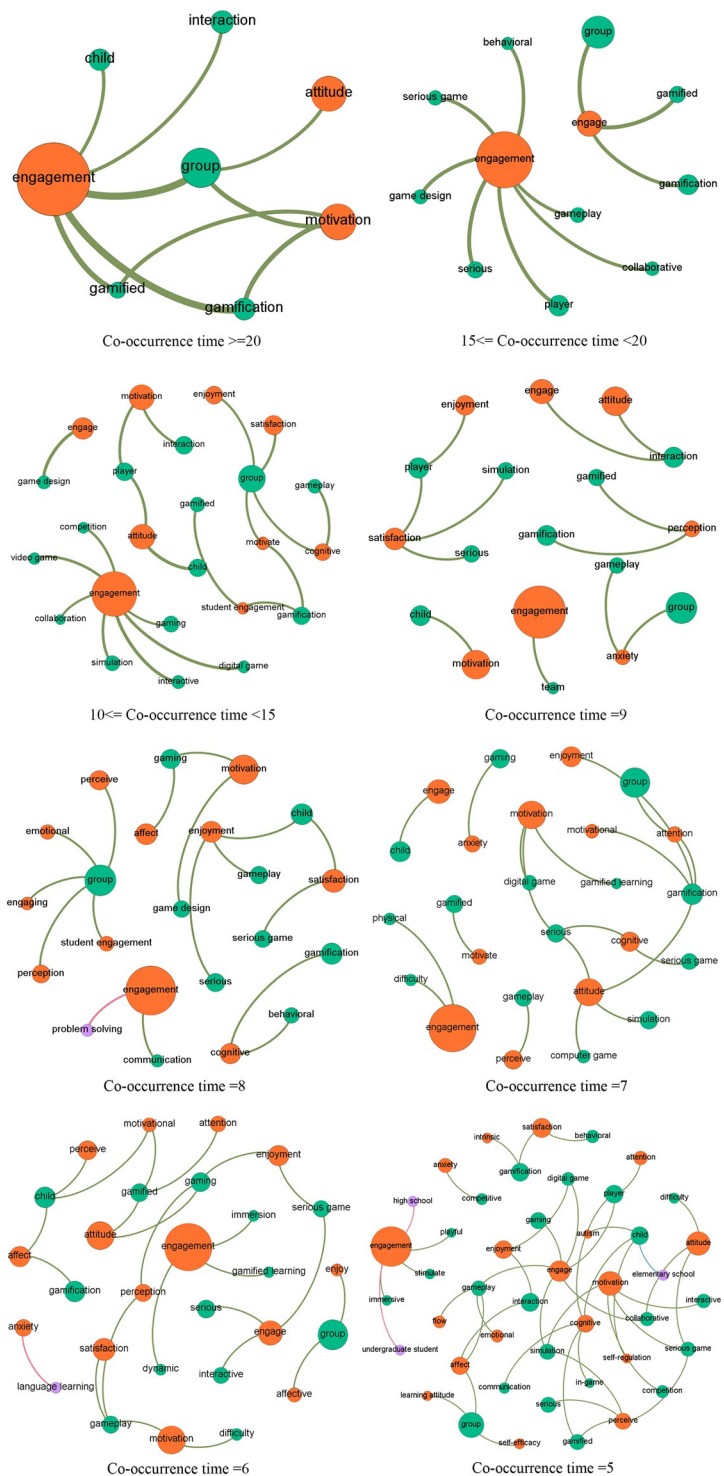

**Fig 6. Relationships between topics with co-occurrence of not less than five.**

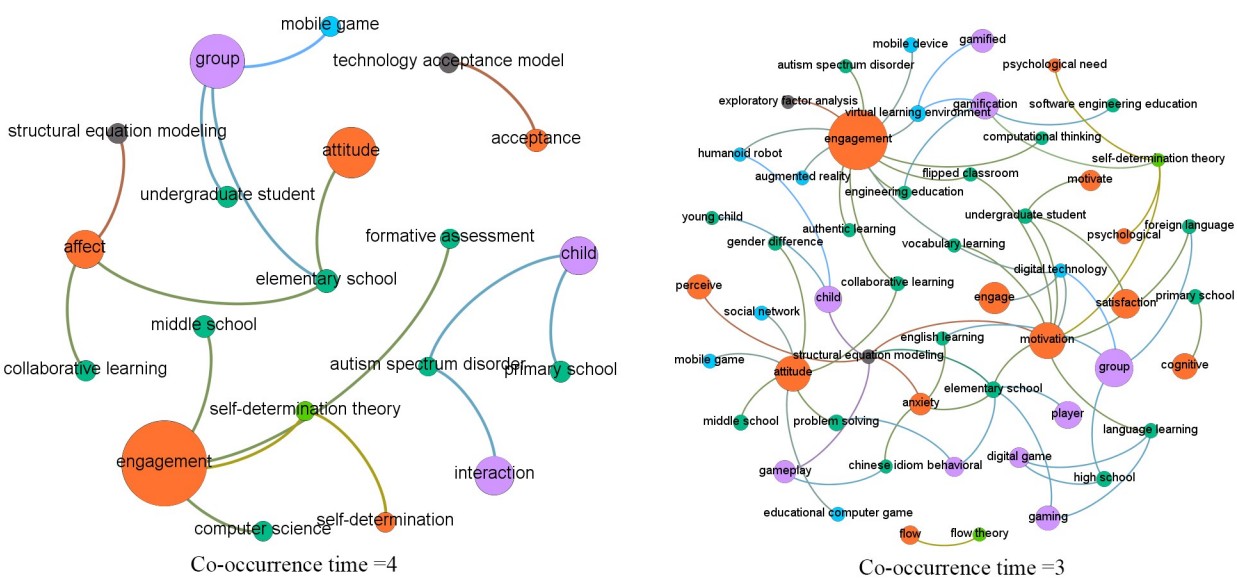

**Fig 7. Relationships between topics with co-occurrence of three and four.**

and researchers' growing awareness of emotions' significance in DGBL also stimulates research into diverse aspects and issues.

**Perceived usefulness/ease.** An essential stream of research on DGBL is learners' acceptance of these systems [26]. Important factors impacting user acceptance of DGBL systems are commonly found to be relevant to perceived ease of use, perceived usefulness, self-efficacy, and satisfaction, which are usually examined using TAM (see Fig 7 for the close relationship between acceptance and TAM), a reliable tool for predicting end-users acceptance or adoption

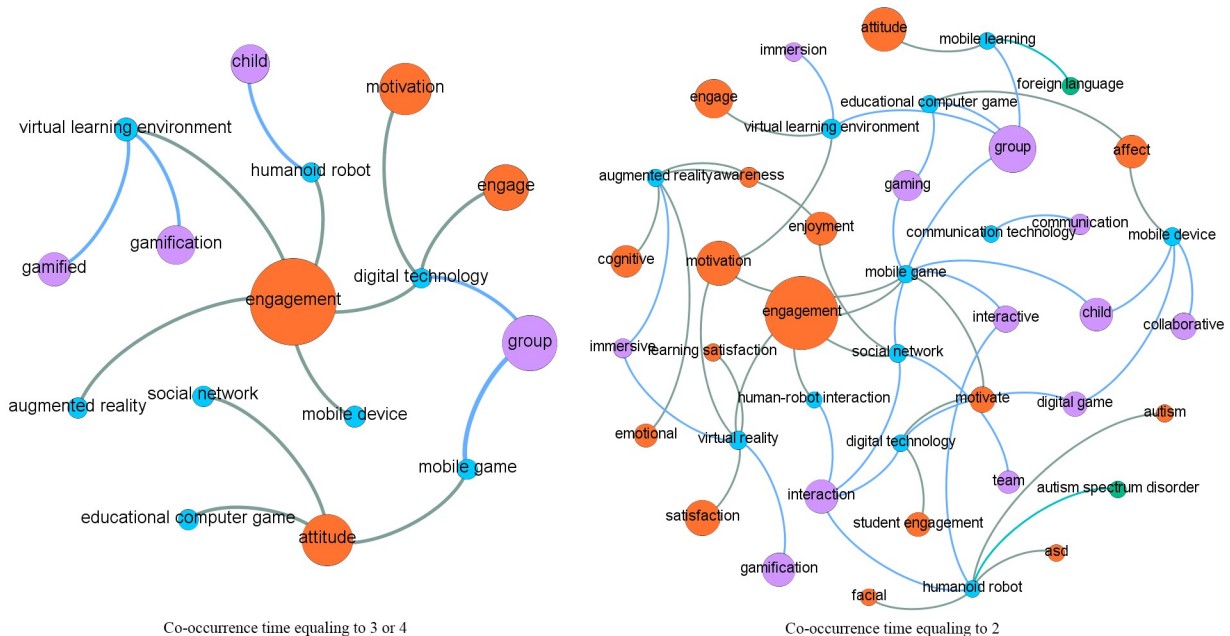

**Fig 8. Relationships between topics (with co-occurrence ranging from two to four) linking with applications/devices.**

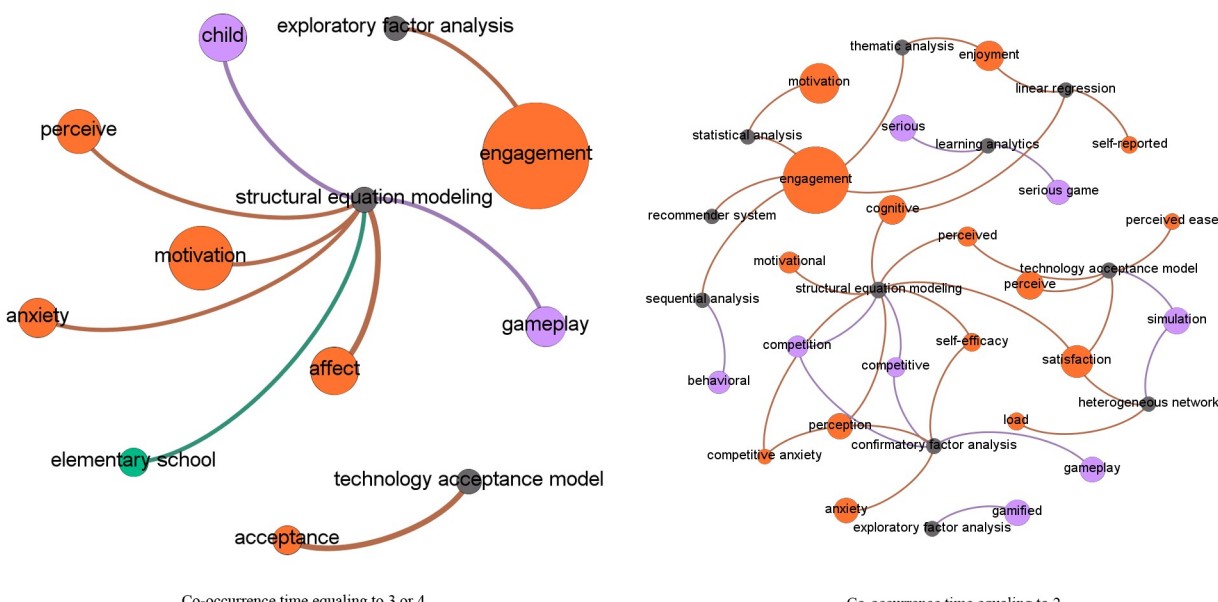

Co-occurrence time equaling to 3 or 4

Co-occurrence time equaling to 2

**Fig 9. Relationships between topics (with co-occurrence ranging from two to four) linking with analytical technologies.**

of new technology. The use of TAM facilitates the understanding of learners' behaviors and intrinsic and extrinsic factors in DGBL system use. For example, in a TAM-based study [27], learners learning with virtual reality games show higher game satisfaction and self-efficacy than those using traditional high-fidelity simulations. This is because learners are more immersed in and keen to keep playing with games, and meanwhile capable of transferring theories into practice with an integrated feedback mechanism helping them evaluate their practice competency. Focusing on children's "continued use of interactive computer game-based visual perception learning systems (p. 76)" in special education, Lin et al.'s TAM analysis [26] indicated a positive effect of learner-instructor and learner-system interactions on children's perceived ease of use and perceived usefulness.

**Socio-emotions.** Emerging technologies with socialization features demonstrate motivational effects to support the basic psychological need for relatedness through promoting social belonging, connectedness, and support (see Fig 7 for the close relationship between social network and mobile game and attitude), which is consistent with the claim that "our relationships are increasingly mediated by technology (p. 6) [28]." Web technologies and social networking sites integrated with DGBL strategies provide enormous opportunities for learners to learn through interacting, communicating, and collaborating socially and virtually [29]. The study of whether and how these digital technologies impact learners' affect and particularly those related to socio-emotions like social support and social anxiety during DGBL is important to understand how to make the best use of socialization functions brought by the technologies during learners' DGBL. In [30], an interactive online game, by translating "evidence-based social-emotional learning strategies into personalized social problem-solving scenes (p. 959)" in virtual worlds, improves elementary-age children's social skills, behavior, satisfaction, and confidence in their social relationships.

**Motion-sensing games.** In motion-sensing games, players are required to control and complete gameplay through body movements [31]. Players' posture and body movements also indicate their affective states [32] during gameplay for the understanding of their emotional experiences. In [33], motion-controlled video games promote learners' information processing

and increase their cognitive resources assigned to task performance, which is mainly due to the games' advantages in keeping learners motivated and engaged during task completion. The increasing interest in studying emotions in motion-sensing games can be a result of the advances in three-dimensional technologies' improved efficiency in capturing body position and movements, which adds value to detecting and understanding learners' affective status during DGBL. However, the coding of body movements remains challenging with a lack of specific frameworks defining basic units of gesture and body movements for emotion detection [34].

**Competitive gameplay anxiety.** Competitive anxiety (see Fig 6 for the close relationship between competitive and anxiety), as an essential inhibitor of task achievement [35], is increasingly studied during gameplay, particularly in competitive DGBL due to its commonness and potential negative impacts on learners' performance (e.g., higher competitive anxiety usually results in lower gameplay enjoyment and engagement). However, competitive anxiety's negative effect on gameplay interest is not conclusive in the literature. Hong et al. [36] found competitive anxiety's positive relationship with interest in gameplay, and they attribute it to the stimulated players' interest and the gameplay control triggered by the higher state anxiety. Hong et al. [37] also suggest competitive anxiety's positivity in eliciting a change from goal-oriented to stimulus-driven attention, thus stimulating players' focus on competitive situations and their intention to play again. Whether competitive anxiety plays a positive or negative role in gameplay also relates to individual personality. Hong et al. indicate a positive correlation between gameplay anxiety and deprivation-type (D-type) epistemic curiosity (EC), which is because D-type EC involves desires for knowledge and worries for increasing working efficiency, thus triggering anxiety to play.

**Location-based games.** With the integration of mobile technologies with location awareness features, location-based learning has been significantly extended from mere content identification to interactive discovery to promote learners' engagement and immersion in contextual activities and enhance their experiences of learning from surroundings and with each other [38]. During such immersive gameplay, learners' immersion is measured in the forms of different levels of cognitive and emotional involvement [39]. It is commonly reported that location-based games improve learners' learning attitudes and achievements by guiding them in finding the target and showing them the learning tasks and materials accordingly.

## Research foci during 2018–2019

In this period, diverse emotions and game-related issues were concerned. This is associated with DGBL's effectiveness being increasingly recognized in diverse contexts and for varied educational purposes. Also, the increasing prevalence of varied learning strategies (e.g., contextual learning, personalized learning, and immersive learning) stimulates researchers to integrate them into DGBL with learners' emotions being considered. Additionally, with the resort to varied technologies like eye-tracking, EEG, electrocardiography (ECG), and machine learning (see Fig 4), researchers start to conduct more fine-grained research with more detailed issues being concerned.

**Subject-specific emotions.** Due to the increasing interest in developing educational games specific for particular subject learning (e.g., game-based writing/vocabulary learning, word game, and code-learning game) and the growing awareness of the importance of learners' affective status in impacting gameplay performance, diverse emotions related to specific subject learning (e.g., English anxiety, writing attitude, and computer anxiety) (see Figs 6 and 7 for the close relationships between anxiety and language learning; anxiety and English learning; anxiety and Chinese idiom) start to emerge in this period. In [40], learners show

significant improvements in Spanish learning enjoyment, engagement, and attitude towards computer science brought by game-like activities integrated into grammar-specific mobile learning applications. Hosseini et al. [41] also suggest that game-based lectures are more engaging and enjoyable, contributing to learners' improvements in learning perception of computer science concepts, engagement, and teamwork. There are also studies showing variations in learning achievement associated with anxiety levels. For example, Cheng and Chen's study [42] indicates that learners with lower English anxiety show higher achievement compared to those showing higher anxiety while using an English mobile learning system integrated with interactive games.

**Affective regulation (strategy).** Self-regulated learning involves the regulation and management of individual learning to comprehend personal performance and achieve learning goals [43]. Self-emotion regulation, as a crucial element in self-regulated learning individuals adopt to enhance, decrease, or sustain specific affective states for outcome achievement [44], is increasingly researched and recognized as important in determining improvements in DGBL. During gameplay, learners showing higher self-regulation abilities tend to experience positive flow affect [45]. Attention has also been paid to how and under what situations self-regulation elicits the most positive effect. An exploratory study [46] indicates the benefits of engaging in effective regulation of cognitive reappraisal and acceptance to reduce frustration and confusion. Chen and Sun [47] suggest enhancing learners' immersion in DGBL by balancing between learners' skill levels and game challenges since learners are more likely to self-regulate their flow state when a difficult choice is given to them.

**Intrinsic need/motivation.** As a socio-emotional factor supporting effective interactions, intrinsic motivation, an "innate psychological need for competence and self-determination (p. 3) [48]," includes learners' interest in their learning subjects and learning processes. Accumulative evidence indicates gamified interventions' varied impacts on learners' learning participation cross intrinsic or extrinsic motivation. Compared to extrinsic motivation, according to self-determination theory, intrinsic motivation increases game enjoyment more. Such a principle is increasingly considered during the implementation of meaningful gamification mechanics with a key goal to construct meaningful and intrinsically learner-centered DGBL experiences. Thus, how to trigger more intrinsic motivation through gamification element use becomes a prevalent research topic. In [49], the use of gamified interactive response systems (IRSs) contributes to greater intrinsic motivation to learn English as compared to those using whiteboards. This is because gamified IRSs are attractive and can create easy delectation of learning atmosphere where instructors plan richer and more interesting courses to promote learners' learning interest and classroom interactions, thus triggering their intrinsic motivation to learn. In addition, when integrated with a feedback mechanism and clear goals [50], educational games more effectively contribute to learners' intrinsic motivation and thus their perceived learning and learning outcomes.

**Emotion dynamics and changes.** According to [51], a broad range of emotions can be triggered during DGBL simultaneously and dynamically. Hence, it is critical to avoid a mere focus on retrospective assessment to understand learners' perception of a current situation; however, this is contradictory to most extant literature retrospectively measuring learners' affective states in DGBL. On this account, scholars start to pay attention to emotional dynamics by measuring changes in emotions during DGBL (see Fig 6 for the close relationship between dynamics and engagement). An autoregressive model [52] indicates a significant relationship between learners' control perception and emotions in every round of gameplay designed for human liver functionality instruction. Therefore, it is essential to determine whether and when an adaptive change in specific design characteristics should be introduced for subsequent game tuning to obtain ideal gameplay experiences and performance.

**Mood/affective disorders.** Children with autism spectrum disorders commonly have difficulties in recognizing facial expressions and emotions (see Fig 7 for the close relationship between child and autism spectrum disorder and Fig 6 for the close relationship between child, autism, and cognitive). A promising solution to resolve such difficulties is to exploit robotic toys' potentials to facilitate intervention processes of autistic children by enabling them to experience diverse social behaviors through interacting with robots. An examination of socio-emotional development [53] indicates humanoid robots' effectiveness to promote high functioning autistic children's acquisition of facial expression recognition skills during gameplay and their engagement in interactions. The increasing interest in researching affect and emotions in children with mood or affective disorders is partially contributed by technological advances with eye-tracking, EEG, ECG, and facial emotion detection technologies being popularly introduced to more conveniently detect emotions compared to traditional data collection methods like questionnaires and interviews requiring self-reporting.

**Impact of human factors on attitude towards personalized DGBL.** There is a trend towards customizing avatars, scenarios, and gaming content tailored to individual learners to promote DGBL immersion (see Fig 3). In [54], learners were highly engaged with customizable avatars that allow them to customize their own unique avatars in terms of gender, appearance, profession, and animal companion in line with their preferences. However, such personalized gamification experiences vary across individuals with different characteristics (e.g., cognitive styles, learning styles, game experiences, and personality traits). Thus, the understanding of how individual characteristics affect gamification experiences is important to provide insights into effective gamified learning intervention design. Chen et al.'s study [54] on the impact of cognitive styles (i.e., Pask's Holism/Serialism) on learners' reactions towards customizable avatars suggests that Serialists and Holists experience higher levels of engagement and sense of presence, respectively.

There are also studies (e.g., [55–57]) examining the impacts of learning styles and personality traits on gaming experiences and perception. Sepehr and Head's study [57] suggests that learners with higher levels of dispositional competition are more intrinsically motivated during gameplay and considered the games more competitive. Focusing on personality traits' impact on learners' perceptions of and engagement with gamified learning interventions, Buckley and Doyle [55] suggest that extraverted individuals, compared to conscientious ones, are motivated more by gamification. Buckley and Doyle also found that emotionally stable individuals, compared to neurotic ones, show better performance, and individuals tending towards global learning favor gamification more, whereas highly conscientious individuals commonly show a negative gamification perception. Buckley and Doyle and Ghaban and Hendley indicate that enhancing learners' motivation cannot guarantee their improvement in learning since varied personalities respond differently to gamification elements. Therefore, for tutoring system designers thinking of integrating gamification, it is crucial to provide learners with gamification elements matching their individual characteristics, and that can be updated dynamically based on their behaviors during gameplay.

**Contextual DGBL.** Contextual learning involves providing simulated contexts to apply knowledge and opportunities to mastery learning through repeated practice and personalized guidance and feedback [58]. Digital games provoke active learner involvement during contextual learning through exploration, experimentation, competition, and collaboration. Learners' immersion in the contextual DGBL activities enhances their learning motivation and interest [59], and has become an increasingly popular practice in healthcare education. In a contextual role-playing game [60], learners learn to save others' lives in varied cases by collecting data concerning normal and abnormal ECG waves and completing relevant testing tasks. Contextual games' effectiveness in enhancing learners' learning motivation, attitudes, and satisfaction

is mainly because integrating learning materials into gaming scenarios allows learners to learn with pleasure and interactively [61], and situating them in real-world simulations enables them to link real-life problems to their learned knowledge [62].

**Collaborative games and team-based gaming.** Compared to individual learning and conventional instructions, collaborative learning is increasingly considered effective to elicit higher achievement concerning individuals' cognitive development [63]. The effectiveness of collaboration strategies can be enhanced by integrating gamification mechanisms that are advantageous in promoting motivation and engagement to complete educational tasks [64] (see Fig 6 for the close relationship between group and attitude, engagement, and motivation). Particularly, in an era where technological advances proliferating in every aspect of social life, designers increasingly focus on developing and designing technological support for collaborations in DGBL activities [65] where learners practice self-regulation and social skills to achieve team goals. Hanghøj et al. [66] indicate the positive effect of collaborative role-playing games in promoting the sense of social belonging among learners with social difficulties by increasing their social participation and engagement through team collaborations. Doumanis et al. [64] also claim that learning in virtual multimodal contexts with integrated collaboration and gamification elements promotes learners' motivation and engagement in accomplishing collaborative tasks by involving learners in intuitive and rich interactions with others and the virtual learning contents.

**Reward mechanisms in DGBL.** Similar to games where rewards play an essential role as motivators and feedbacks, in education, rewards are used to motivate and engage learners and recognize their efforts. The effectiveness of rewards as a form of either identified regulation (learners' consciously-valued rewards like tokens or badges) or integrated regulation (rewards aligning with learners' goals in enjoying gameplay) is commonly recognized [67]. However, in practice, rewards have varied effects on gameplay, especially those showing no positive impacts on learners' game experiences, thereby limiting their values to learners. Effective rewards are expected to facilitate positive game experiences by motivating learners to actively participate in learning tasks to obtain the rewards [68]. Also, prompting rewards from gamified systems are informational feedback on learners' short-term performance, which can promote their sense of achievement [69] and encourage them to set new goals and concentrate on gaming progresses [70]. The variations in different rewards' effects on DGBL and the significance of effective rewards stimulate scholars' interest in comparing different types of reward mechanisms and their effects on learners' gameplay. A reward strategy comparison during gamified instruction [71] indicates that forfeit-or-prize and prize-only strategies stimulate learners' learning motivation. This is because learners experiencing the forfeit-or-prize strategy tend to experience higher levels of anxiety. Therefore, it is necessary to balance between rewards and learning anxiety (pressure, stress, or competition) via reward strategy design to boost motivation but suppress pressure. In [69], performance-contingent and completion-contingent rewards were compared. With an example of English vocabulary learning via arrow-shooting gaming, the authors suggest that both performance- and completion-contingent rewards facilitate learning and motivation. The higher perceived competence and autonomy brought by the performance-contingent rewards is due to the rewards' success in delivering sufficient competence information, which, according to cognitive evaluation theory [48], negates the controlling effect, thereby positively influencing learners' motivation.

## Discussion

Based on the thematic evolution and social network analyses, this study explores important topics covered within 393 DGBL research studies in which learners' affective status is

considered. Consistent with [8] highlighting diverse types of emotions (e.g., satisfaction, engagement, and attitude) in virtual learning, we more specifically identified detailed aspects of concepts such as emotions related to specific subject learning (e.g., computer anxiety and English anxiety) and social emotions. Our results similar to [72] in revealing gaming features' important roles in affecting learners' engagement in DGBL. However, their only focused on engagement, while we examine a wider range of concepts. Thus, our conclusion about game features' impacts do not limit to engagement but on varied emotions such as satisfaction and anxiety. We also summarized the important topics detected in our findings into a research framework and further proposed suggestions for future research on learners' affective status in DGBL.

## Research framework and implications for future research

Based on our analyses, by following previous frameworks concerning digital game types [73], emotion-related concepts [74], applications and devices [17], and analytical technologies, we provide a framework (see Fig 10) concerning research on learners' affective status in DGBL. Additionally, by examining the top empirical studies measured by annual citations in each sub-period (see S1 Table), suggestions for future research are provided.

**Affective states and socio-emotional factors.** In terms of affective states and socio-emotional factors, four dimensions (i.e., emotions, mood, personality traits, and socio-emotional factors) are used to categorize popular concepts appearing in the corpus. Referring to evolution analysis, overall, affective state emotions and socio-emotional factors begin to be considered in DGBL in an early stage, particularly satisfaction, motivation, engagement, and encouragement. Comparatively, affective state mood (e.g., confusion, frustration, anxiety, and

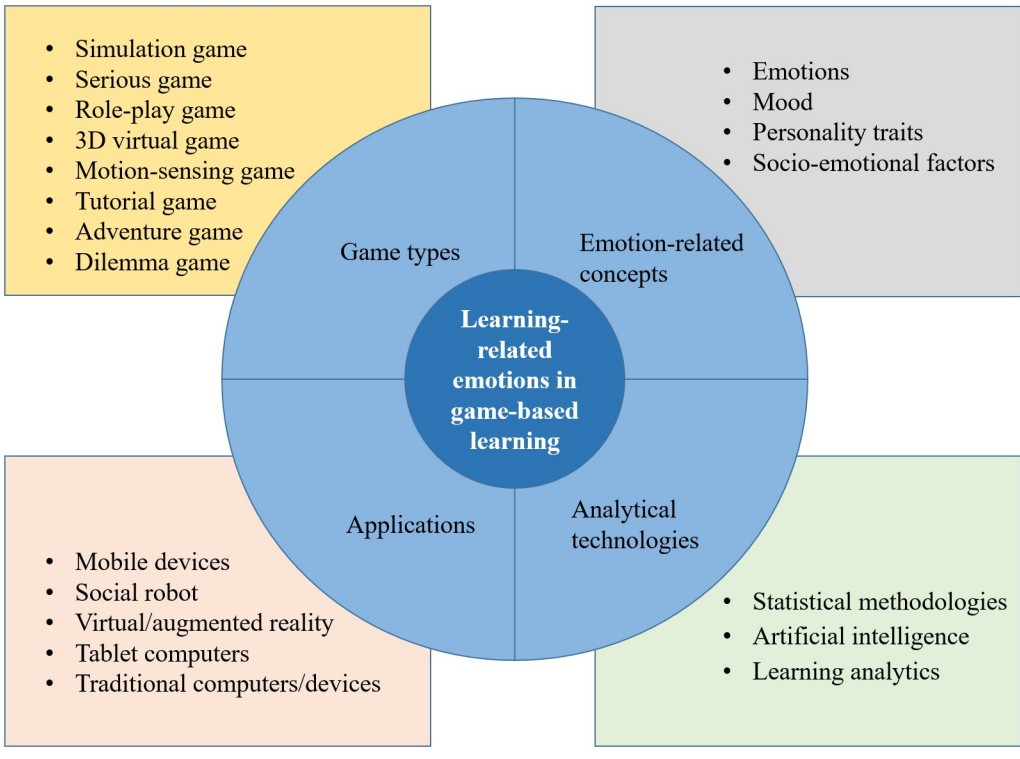

**Fig 10. Framework of important topics in research on learners' affective states in DGBL from the perspectives of affective states, digital games, applications and devices, and analytical technologies.**

enjoyment/joy) and personality traits (e.g., attitude, aggressiveness, and emotional stability) appear in the latter stage of DGBL research. Though the volume of literature concerning learners' affective status in DGBL is increasing, there is a paucity of longitudinal studies. As it is common that once the novelty of instructional strategies like DGBL wears off, learners would be fatigued and bored, thereby eroding their engagement and motivation. Hence, as suggested in several studies (e.g., [60, 75]), it is worthwhile to conduct longitudinal studies to evaluate gamification's long-term effects on learners' positive learning emotions. For example, since it was the first time for participants in most studies to participate in DGBL, the investigation of the engaging impacts of gaming activities when they are implemented regularly and become less novel [76] appears essential.

As discussed before, learners differ in demographic factors (age, gender, education levels, and cultural contexts) and personality traits (competitiveness orientations, spatial ability, and learning and cognitive styles), which may impact gamification' effect on affect and emotions (e.g., [60, 77]). Hence, the influence of these variables ought to be assessed (e.g., how competitiveness engages or enables learners' problematic behaviors) [78] to gain a holistic and contextual understanding of DGBL design and implementations specific for certain learning contexts and different groups of learners. Also, varied educational conditions (e.g., study subjects and instructional strategies) under which gamification more positively affects individual learners (e.g., [79, 80]) should be examined and compared.

**Types and design of games.** We identified different types of digital games, for example, simulation games, serious games, role-playing games, and motion-sensing games, where learners' affective status is considered. However, as most studies examined gamification's effects as an overarching concept, future studies are advised to evaluate gamification's specific elements to parse out different mechanics' effectiveness [25, 81]. For example, in gamified system design, researchers are suggested to make the most of digital gaming mechanics that are best employed using computers and virtual worlds [25]. Also, as greater numbers of gamification mechanics cannot necessarily contribute to better learning performance [82], the optimal number of badges maintaining active engagement ought to be determined [76]. The isolation of specific game mechanics and assessment of their effects on learners' affect and emotions enables a better understanding of nuanced and effective gamified learning intervention design to maintain and promote motivation and engagement.

According to Hwang and Wang [83] and Calvo-Ferrer [84], a clear goal with a moderately high cognitive load is more likely to facilitate effective learning. It is thus important to consider challenging learners' thinking skills and integrating clear learning objectives into game dynamics during game design. Also, to enhance enjoyment and decrease anxiety, cognitive and metacognitive scaffolding are important to foster high levels of subjective control [84]. Wei et al. [85] and Wu and Huang [86] also suggest the contribution of assistance tools and strategies to promote learners' motivation and engagement by constructing a low-anxiety learning atmosphere and enhancing learners' immersion and intrinsic interest. It is thus essential to consider integrating personalized assistance and providing scaffolding in educational games to release or moderate anxiety to thus promote game performance. Game developers and designers can employ open-source models and stimulate learners' to customize games via source code manipulation [87]. This would provide more personalized learning experiences for learners to better engage in game activities, thus promoting learning and game performance. In addition, it is essential to integrate grouping rules and tactics to achieve effective heterogeneous collaborations, thereby contributing to peer interaction in real time.

**Applications and platforms.** Our analyses reveal that traditional computers and devices are popularly adopted to support DGBL, which is mainly because game systems are commonly developed based on existing packages or systems to save costs in human resources and time

[88]. With the advances in information technologies in wearable and handheld devices, there is an increase in educational games to be supported by mobiles, tablet computers, and virtual/ augmented reality. To maintain learner engagement and ensure enjoyment while concentrating on solutions and resolving challenges during gameplay, handheld devices are effective by providing individual learners with equal opportunities to participate in the games and interact with others. Therefore, effective ways to arrange learning content and materials in handheld and mobile devices are expected to provide learners with prompt peer feedback and equal learning opportunities [89] to trigger positive emotions during DGBL, thus promoting gameplay and learning performance and outcome. Another promising stream of research lies in the wearable learning technologies with personalized data to facilitate the understanding of individuals' affective status during gameplay to allow more personalized learning experiences. This is in line with the definition of "wearable personal learning technologies [90]," involving gathering data from learners wearing the devices and the surroundings to promote instruction differentiation and learner engagement.

**Analytical technologies.**   Increasingly diverse technologies are used to facilitate the understanding of learners' affect and emotions during DGBL. Traditional statistical methodologies (e.g., Bayesian network, SEM, regression analysis, survival analysis, and sequential analysis) are mainly employed to analyze structured data collected using traditional questionnaires and survey data after gameplay. However, more attention should be paid to how to directly capture data concerning learners' activities and emotions during gameplay. A potential methodology is by combining questionnaires with game log and assessment data and by increasing measurement robustness [91]. Additionally, techniques like "think aloud" during gameplay, video analysis, click-data, and psycho-physiological measures like EEG, ECG, and skin conductance, are expected to gauge learners' state-to-state emotional experiences during playing [59] to infer their learning status (e.g., emotions, cognition, and attention). Alongside technological advances (e.g., Internet of Things, wearable and cloud technologies) is the realization of collecting learning data at high-frequency, fine-grained, and micro-levels in a real-time mode. These multimodal data with great diversity can be analyzed using affective computing and learning analytics, a powerful tool for developing adaptive educational technologies [92], to facilitate the understanding of learners' trajectories in the games and explore their relations to learners' self-reported goals and purposes and outcomes like engagement [93]. A promising direction is multimodal learning analytics, which detects, incorporates, and analyses learning traces obtained from varied sources/channels and on a large scale to enable a comprehensive understanding of learning processes. The analysis results on affect and emotions can be combined with individual learners' preferences, interests, and goals to facilitate the provision of personalized support and interventions via recommender systems like collaborative filtering technologies to promote their engagement in DGBL.

## Conclusion

This is the first in-depth study to track advances in research focusing on learners' affective states in DGBL using thematic evolution and social network analyses. Such timely work is needed with emotions and affect becoming increasingly important in DGBL. Results reveal that in addition to common affect like satisfaction, engagement, attitude, and motivation, diverse types and aspects of emotions like subject-specific emotions, affective regulation, emotional dynamics, intrinsic need, and mood/affective disorders are increasingly researched. Alongside varied emotions being concerned is the increasingly diverse types and aspects of games, for example, personalized DGBL, contextual games, and gaming reward mechanisms. The increasingly active research on emotions in DGBL attributes greatly to the introduction

and implementation of fast-developed and innovative applications and devices like mobile and networking technologies, robotics, virtual/augmented reality, and advanced analytical technologies like learning analytics, affective computing, and machine learning.

It is worth highlighting the framework proposed based on the analyses from the perspectives of affective states, game types, game design, and applications and devices supporting DGBL, and analytical technologies, which depicts the mainstream topics in this increasingly active field of research. Such a framework is potential to guide future research in the field, and we accordingly provide suggestions for future research on learners' affective states in DGBL, including 1) children's anxiety, attitude, engagement in collaborative gameplay integrated into humanoid robots, 2) individual personalities and characteristics' influence on attitude or motivation towards diverse games (e.g., location-based/context-aware games and virtual/augmented reality games) to inform personalized support, 3) emotion dynamics and changes in DGBL, 4) multimodal data analyzed using learning analytics and advanced machine learning algorithms to facilitate a deeper understanding of affective states during gameplay, 5) allowing learners to customize games to promote engagement, 6) balance between learners' skill levels and game challenge to promote immersion, and 7) balance between rewards and learning anxiety through effective reward strategy design.

In future work, it will be interesting to consider conducting correlation analysis with statistical tests to explore the relationships between topics in statistics. Second, we focus on using typical approaches to bibliometric analysis, where hypothesis testing is not needed. Nevertheless, we believe it would be interesting to come out with effective ways to integrate hypothesis testing into traditional bibliometric analysis for more robust analysis on large-scale literature data.

## Supporting information

**S1 Table. Top studies published in three periods.**
(DOCX)

**S1 Dataset.**
(XLSX)

## Author Contributions

**Conceptualization:** Xieling Chen, Di Zou, Haoran Xie, Gary Cheng.

**Data curation:** Xieling Chen, Di Zou, Lucas Kohnke.

**Formal analysis:** Xieling Chen, Di Zou.

**Funding acquisition:** Lucas Kohnke.

**Investigation:** Xieling Chen, Di Zou.

**Methodology:** Xieling Chen, Di Zou, Haoran Xie, Gary Cheng.

**Project administration:** Lucas Kohnke, Haoran Xie.

**Resources:** Lucas Kohnke, Haoran Xie.

**Supervision:** Haoran Xie, Gary Cheng.

**Validation:** Gary Cheng.

**Visualization:** Lucas Kohnke, Gary Cheng.

**Writing – original draft:** Xieling Chen, Di Zou.

**Writing – review & editing:** Xieling Chen, Di Zou.

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
