## [Decision Letter · Decision Letter 0]

27 Apr 2021

PONE-D-21-09587

Aﬀective states in digital game-based learning: thematic evolution and social network analysis

PLOS ONE

Dear Dr. Zou,

Thank you for submitting your manuscript to PLOS ONE. After careful consideration, we feel that it has merit but does not fully meet PLOS ONE’s publication criteria as it currently stands. Therefore, we invite you to submit a revised version of the manuscript that addresses the points raised during the review process. I have invited experts in the field and they have raised concerns regarding the methods,In particular, the choice of the methods should better explained, grounded in literature and allow a reproducible research.

We look forward to receiving your revised manuscript.

Kind regards,

Mohammed Saqr, Ph.D

Academic Editor

PLOS ONE

Journal Requirements:

2.  Please note that board membership at PLOS should be declared in your competing interest statement.

Reviewers' comments:

Reviewer's Responses to Questions

**Comments to the Author**

1. Is the manuscript technically sound, and do the data support the conclusions?

Reviewer #1: Partly

Reviewer #2: Partly

2. Has the statistical analysis been performed appropriately and rigorously? 

Reviewer #1: No

Reviewer #2: I Don't Know

3. Have the authors made all data underlying the findings in their manuscript fully available?

Reviewer #1: No

Reviewer #2: Yes

4. Is the manuscript presented in an intelligible fashion and written in standard English?

Reviewer #1: No

Reviewer #2: Yes

5. Review Comments to the Author

Reviewer #1: I. Summary of the research and overall impression

Title: ‘Affective states in digital game-based learning: thematic evolution and social network analysis’

I’d like to thank the authors for their interesting submission. The study the manuscript reports on addresses an important aspect of learning, i.e., virtual learning that is nowadays related to DGBL. The goal is to synthesize how emotions are addressed in DGBL literature. The paper provides an overview of learners’ affective status in DGBL based on 393 research articles. In principle, this article could be of interest to the general readership of the Plos One Journal. However, there are some major issues that need to be addressed.

• The authors should better explain why their review is important, in light of previous research and of questions yet to be answered within the field.

• They should make a substantial, complete and exhaustive literature review (i.e., theoretical background).

• Based on this previous literature, the authors could formulate hypothesis related to the links that are expected between emotion and DGBL.

• Also the scope of the study is not always clear. For instance, the aim of the paper is to link emotions and DGBL, but within the results we can see that the links between autism - which is not an emotion - and DGBL are also tested.

• The method section should be developed more (e.g., statistical analysis).

• With a complete theoretical background (and maybe some hypothesis clearly formulated), the authors would be able to indicate which results align with previous literature and to explain why other do not correspond to what was found in previous studies.

• The authors need to rework their manuscript in order to put the different elements where they belong.

II. Specific comments

a) Abstract

• The problem statement should be made more explicit.

• ‘learners’ emotions is increasingly important in understanding learners’ learning experiences within digital game-based learning (DGBL)’ It would be useful to explain why and how it is important (one or two sentences).

• The authors summarize the main research question and key findings but the abstract is quite long and could beneficiate from deleting some unnecessary details.

b) Introduction

• The links between paragraphs are not always clear to me (e.g., between ‘Understanding students’ affective status during learning’ and ‘evidence show digital games potentials to nurture and sustain high levels of learning motivation and engagement’). I have the impression that sometime, steps are missing between the ideas that are developed.

• It would be helpful to clearly state from the start that the study is limited to virtual learning.

• Also, on all students or limited to high-school students, university students …?

• It would be helpful to give a clear definition of what are ‘topics’ for DGBL

• The ‘what’ is addressed: ‘The goal is to synthesize how emotions are addressed in DGBL literature’

• The ‘why’ is partially missing: it is not clear from the introduction, but I think that working on links between paragraphs might help better explaining why the paper is important? The authors should explain why their study matters and put the research in context.

• The ‘how’ is missing : The authors should briefly present how they intend to answer their research questions

• The way the manuscript is organized is missing

c) Literature review (i.e., theoretical background)

• What is contextual focus (row 86)?

• Engagement is an emotion? I don’t quite understand the links between rows 83-86 and 87-91?

• ‘For example, a close correlation between an emotion-related topic A and an analytical technology-related topic B means that B is popularly adopted to study issues related to A’. Could you give a concrete example? This might help understanding better why your review is important.

• The literature review is too narrow. I think the authors should deepen what’s have been done in previous research and that legitimates their research

• ‘We divided the whole period into three sub-periods each consisting of two years’: why this particular division?

• Rows 114-128: why within the literature review section?

d) Methods

• The emotions, educational games, gaming approaches, methodologies, concepts and different types of platforms supporting DGBL should be defined (within a theoretical section instead of the method section?)

• Search query: the authors have to specify on what search engine(s). If I understand correctly, the search queries are combinations of the different terms? (e.g., emotion+game+learn; affect+game+learn …). It should be explained better.

• Natural Language Toolkit: the authors should explain it in a few sentences.

• The topics in four categories (i.e., digital games, affective status, applications and devices supporting DGBL, and analytical technologies): the authors should explain from where the four categories come: from previous literature or were they identified by them (based on the 393 research articles)?

• Rows 183-198: within a theoretical section (or the literature review section) instead of the method section?

• Before speaking about nodes and their size, the authors should introduce SNA and define what a network is, then explain how they use SNA in the context of their research.

• It is not very clear (stated) that SNA is also applied on emotions. But I think that it is because the term ‘topic(s)’ is used to represent different things, and as reader, I’m not always sure of what topic(s) refer(s).

e) Results

• Thematic correlation: too poorly developed, the reader has to look at the figures (6 to 9) to (try) understanding the results.

o The results are explained later within the discussion section � they have to be removed within the results section.

• Also, some terms appear in some figures, but I don’t understand why, since it was not explained earlier that these terms would be tested in the correlations (e.g., elementary school, high school, flow, engineering education, autism …)

f) Discussion and conclusions

• Rows 271-272: ‘we more specifically identified detailed aspects of concepts such as emotions related to specific subject learning (e.g., computer anxiety and English anxiety) and social emotions’. Unless I’m wrong, it was not previously indicated that emotions related to subject learning would be investigated?

• The major problem encountered within the discussion is that since the literature review was poorly documented, and since no hypothesis formulated for the expected correlations between constructs, the reader is not able to know if the results support previous literature, if they bring something new to the field, support the conclusions …

• I’m also under the impression that a substantial part of the literature review (i.e., theoretical background) - which needs to be addressed in the beginning of the paper - is included within the discussion.

• Again, scope of the study?? ‘Mood/affective disorders’ section?

• If I’m correct, the authors do not discuss any limitations of their study.

g) Statistical analysis

• It would be nice to see some data descriptive analyses (e.g., a table with the means, standard deviation, median, Kurtosis and Skewness …)

• Between what and what exactly were computed the correlations? The co-occurrences? Why working on the co-occurrences and not the occurrences? The authors should explain how they proceeded to the correlation analysis within the method section.

• For the correlations, did the authors use Pearson? Spearman? Did they verify the normality of the data in order to choose the more reliable coefficient?

• Are the correlations significant? The authors need a table with the p-values for each correlation set.

h) Data and supporting information

• I’m not entirely sure that the data provide enough evidence for the authors’ conclusions

• The authors did not provide a sufficient amount of data and information for other researchers to recreate the analyses

i) Figures and tables

• Figures 2 to 5: too dense and small, it is difficult to read. Also, for figure 3, two topics are represented within the same figure. Why? Because it is a little bit confusing.

j) Minor issues

• Make sure using same terms to not confuse the reader.

e.g., (abstract): ‘Specifically, thematic evolution analysis was conducted to explore topic evolution concerning four different groups (i.e., games, emotions, applications, and analytical techniques) in the corpus. Social network analysis was used to explore the relation between topics to identify their correlations’: if I understood correctly, groups = topics? If yes, authors should use the same term. If no, it should be explained better.

• The sentences are quite long and difficult to understand (e.g., ‘With a focus on learners’ engagement during DGBL, Abdul Jabbar and Felicia systematically analyzed game design features promoting engagement and learning in DGBL based on 91 articles during 2003–2013, identifying complex gaming elements impacting gameplay engagement and learning experiences in DGBL at both cognitive and emotional levels’)

• Present and past tenses : not always used correctly (e.g., for the research questions)

• The article is not presented in an intelligible fashion nor is written in standard English. Please let the manuscript be proofread before resubmission.

• References: APA style? e.g.:

Halloluwa T, Vyas D, Usoof H, Hewagamage KP. Gamification for development: a case of collaborative learning in Sri Lankan primary schools. Pers Ubiquitous Comput. 2018; 22(2):391–407.

Instead of: Halloluwa, T., Vyas, D., Usoof, H., & Hewagamage, K. P. (2018). Gamification for development: a case of collaborative learning in Sri Lankan primary schools. Personal and Ubiquitous Computing, 22(2), 391-407.

Reviewer #2: The authors present a thematic evolution of DGBL research using SNA methods. It is an interesting article that examines the increasingly central topic of educational games. Moreover, the authors highlight some of the most relevant areas for further research.

I do have, however, some concerns regarding the methods of the article.

1. The authors used the code "TS" from Web of Science for the search. This term includes Title, Abstract, Author keywords, and Keywords+. The latter are automatically added by the database. This is not a problem since the authors manually checked the articles. However, it is not clear if for the analysis they used only Author keywords, also Keywords+, or also Abstract or Title terms.

2. The authors should describe in further detail how the DGBL-related terms for the search were selected. How did they come up with "affect", "feeling", "anxiety", etc.?

3. The authors should provide more information of the search to allow reproducibility of results, such as if they analyzed only journal articles, or also conference papers, book chapters, etc. They should also state if they filtered by year, by language, etc.

3. The authors should describe if and how keywords were combined or cleaned before the analysis. For instance, combining singular and plural keywords is very common, as well as synonyms. However, in Fig. 6, for instance, words like engage and engagement appear separately. Do the authors think they could/should be combined to better capture the relation between terms?

4. In Figs. 2, 3, 4, the author should specify how the graphs were constructed. A legend should indicate what the size of the circles represent (number of ocurrences of each keyword, I assume) and the color. Also, the authors need to explain the criteria for adding a keyword under a specific year. Was there a specific threshold for the keyword frequency in order it to consider it popular for a given year?

5. In Figs. 6, 7, 8, 9, why were co-ocurrences of 2, 3 or 4 specifically plotted? In my opinion, an overview of the whole network would be very interesting and even more so if a yearly network was constructed.

6. The authors should explicitly point to the results when discussing the findings of the article. For example, when talking about the co-ocurrence of topics, they should point towards the figure that allows them to establish connections among terms. The authors should discuss the meaning of the different colors representing the different groups of keywords that are commonly together as per the co-ocurrence network.

6. PLOS authors have the option to publish the peer review history of their article (what does this mean?). If published, this will include your full peer review and any attached files.

Reviewer #1: No

Reviewer #2: No

---

## [Author Response · Author response to Decision Letter 0]

26 Jun 2021

Please refer to the attached response to comments.

---

## [Decision Letter · Decision Letter 1]

12 Jul 2021

Aﬀective states in digital game-based learning: thematic evolution and social network analysis

PONE-D-21-09587R1

Dear Dr. Zou,

We’re pleased to inform you that your manuscript has been judged scientifically suitable for publication and will be formally accepted for publication once it meets all outstanding technical requirements.

Kind regards,

Mohammed Saqr, Ph.D

Academic Editor

PLOS ONE

Additional Editor Comments (optional):

Reviewers' comments:

Reviewer's Responses to Questions

**Comments to the Author**

1. If the authors have adequately addressed your comments raised in a previous round of review and you feel that this manuscript is now acceptable for publication, you may indicate that here to bypass the “Comments to the Author” section, enter your conflict of interest statement in the “Confidential to Editor” section, and submit your "Accept" recommendation.

Reviewer #2: All comments have been addressed

2. Is the manuscript technically sound, and do the data support the conclusions?

Reviewer #2: Yes

3. Has the statistical analysis been performed appropriately and rigorously? 

Reviewer #2: Yes

4. Have the authors made all data underlying the findings in their manuscript fully available?

Reviewer #2: Yes

5. Is the manuscript presented in an intelligible fashion and written in standard English?

Reviewer #2: Yes

6. Review Comments to the Author

Reviewer #2: The authors have addressed most of my comments. I believe the quality of the manuscript has improved considerably and it can be published in its current form.

7. PLOS authors have the option to publish the peer review history of their article (what does this mean?). If published, this will include your full peer review and any attached files.

Reviewer #2: No

---

## [Editor Report · Acceptance letter]

19 Jul 2021

PONE-D-21-09587R1 

Aﬀective states in digital game-based learning: thematic evolution and social network analysis 

Dear Dr. Zou:

I'm pleased to inform you that your manuscript has been deemed suitable for publication in PLOS ONE. Congratulations! Your manuscript is now with our production department. 

Kind regards, 

on behalf of

Dr. Mohammed Saqr 

Academic Editor

PLOS ONE